# Latent Preference Coding: Aligning Large Language Models via Discrete Latent Codes

## Abstract

Large language models (LLMs) have achieved remarkable success, yet aligning their generations with human preferences remains a critical challenge. Existing approaches to preference modeling often rely on an explicit or implicit reward function, overlooking the intricate and multifaceted nature of human preferences that may encompass conflicting factors across diverse tasks and populations. To address this limitation, we introduce Latent Preference Coding (LPC), a novel framework that models the implicit factors as well as their combinations behind holistic preferences using discrete latent codes. LPC seamlessly integrates with various offline alignment algorithms, automatically inferring the underlying factors and their importance from data without relying on pre-defined reward functions and hand-crafted combination weights. Extensive experiments on multiple benchmarks demonstrate that LPC consistently improves upon three alignment algorithms (DPO, SimPO, and IPO) using three base models (Mistral-7B, Llama3-8B, and Llama3-8B-Instruct). Furthermore, deeper analysis reveals that the learned latent codes effectively capture the differences in the distribution of human preferences and significantly enhance the robustness of alignment against noise in data. By providing a unified representation for the multifarious preference factors, LPC paves the way towards developing more robust and versatile alignment techniques for the responsible deployment of powerful LLMs.

## 1 Introduction

Alignment has emerged as a key step in the development of large language models (LLMs) (Ouyang et al., 2022; Bai et al., 2022; Touvron et al., 2023; Dubey et al., 2024). The goal of alignment is to leverage human feedback to gauge the generative distributions of LLMs, steering their outputs to be helpful, honest, and harmless (Askell et al., 2021). To this end, human annotators are tasked with expressing preferences among human-curated or machine-generated texts, and these preference annotations serve as supervision signals to further optimize LLMs. Amid the surge in alignment research, significant attention has been focused on optimization objectives, considering both online (Schulman et al., 2017; Munos et al., 2023; Calandriello et al., 2024; Yang et al., 2024b) and offline (Rafailov et al., 2024; Zhao et al., 2023; Azar et al., 2024; Meng et al., 2024; Tang et al., 2024) environments as well as different types of preference annotations, such as scalar ratings (Richemond et al., 2024) and pairwise rankings (Rafailov et al., 2024). Optimization with the well-designed objectives has been widely validated for effectively reducing toxicity (Dai et al., 2024) while significantly improving the truthfulness and coherence of LLM outputs (Touvron et al., 2023). In this work, we study the alignment of LLMs from a different perspective: *Can we exploit the supervision signals more effectively through fine-grained modeling of complex human preference?*

The common practice for preference modeling typically involves estimating a single reward function from human annotations as a proxy for human preference (Schulman et al., 2017; Gulcehre et al., 2023). Recently, human annotations have also been used directly as supervision signals in optimization (Rafailov et al., 2024). These approaches, however, often overlook the challenges in preference modeling that arise from the inherent complexity of human preference (Casper et al., 2023): **(1) Human preference may hinge on multiple factors.** The multifaceted factors entailed by a prompt may not be easily represented by a single reward function, especially when some factors conflict with one another. A typical example is the divergence between "helpfuness" and "safty," which differ dramatically in their preferred response patterns, making it difficult for a single reward

model to achieve the best of both worlds (Mu et al., 2024). **(2) The factors may vary across tasks and populations.** There lacks a unified way to represent all factors. For instance, in text generation, pivotal factors that influence human preference may include informativeness, adherence to length constraints, diversity of expressions, etc. In contrast, when solving math problems, correctness of answers, rigor of reasoning, and clarity and conciseness of solutions could be more dominant. **(3) Accurately determining the relative weights of factors for a prompt is challenging, even if the factors are well-defined.** This is particularly significant when the weights are sensitive to nuances in prompt expression. For example, the prompt "how can I kill a *Python process*" demands less consideration on safety than "how can I kill *someone*" despite similar superficial phrasing.

In light of the challenges in preference modeling, we aim to develop a unified framework for capturing the intricate nature of human preferences, with the goal of achieving (1) the framework can broadly represent human preferences across diverse tasks; (2) the framework allows for automatic learning of preference representations without the need for pre-defined sub-rewards and hand-crafted weights that are required by many existing approaches (Zhou et al., 2024; Rame et al., 2024; Yang et al., 2024c); and (3) the framework is generally applicable to various alignment algorithms and can effectively and consistently enhance their performance.

To this end, we propose Latent Preference Coding (LPC), a novel framework that captures the multi-faceted nature of human preferences through discrete latent codes. LPC introduces a discrete latent space where each code represents an underlying factor influencing holistic preferences. Through variational inference, LPC estimates the latent codes from data, and learns both a prior network and a posterior network. The posterior network infers weights of the latent codes from observed preference annotations, while the prior network is trained to predict the inferred weights based on the input prompt. Together, the latent codes and the predicted combination weights form a mixture of factors that represent prompt-specified human preferences, guiding the generation of completions in LLMs[1]. More importantly, the formulation of LPC is general, allowing for integration with a wide range of offline preference algorithms, including DPO (Rafailov et al., 2024), SimPO (Meng et al., 2024), IPO (Azar et al., 2024), and others.

We conduct extensive experiments to assess LPC across diverse downstream tasks, employing Mistral-7B (Jiang et al., 2023), Llama3-8B, and Llama3-8B-Instruct (Dubey et al., 2024) as base LLMs, paired with DPO, SimPO, and IPO as alignment algorithms. Evaluation results indicate that LPC consistently improves LLM performance across various combinations of base models and alignment algorithms. More interestingly, further analysis over the learned latent codes reveals that LPC effectively captures the underlying distribution of human preferences collected from different data sources, and exhibits robustness against noisy annotations. These results confirm that LPC provides a unified approach for representing the complex structures underlying human preferences and is readily applicable to a wide range of existing alignment algorithms.

Our contributions are threefold: (1) We identify the critical challenge of modeling complex human preferences in LLM alignment and propose Latent Preference Coding (LPC) to address the challenge through discrete latent variables; (2) We derive a tailored optimization objective under the LPC framework, which can seamlessly integrate with and enhance the performance of various offline preference learning algorithms; and (3) Extensive experiments on multiple benchmarks, using various base LLMs and alignment algorithms, validate the consistent effectiveness of LPC over the vanilla counterparts.

## 2 RELATED WORK

### 2.1 LATENT VARIABLE MODELS

The inherent complexity of natural language has motivated the employment of latent variable models in natural language generation (NLG) tasks. These models capture the language characteristics by learning latent variables that govern the generation process. A crucial aspect is the specification of the posterior distribution over the latent variables. Continuous distributions, such as the

---

[1]LPC facilitates the automatic learning of prompt-specific preference representations. On the other hand, human preferences are also shaped by differences across populations. While extending LPC to account for population differences is feasible, it falls outside the scope of this paper. We leave the exploration of personalized LPC for future work.

Gaussian distribution used in the variational auto-encoder (VAE) framework (Kingma, 2013), have been widely adopted for modeling response diversity (Zhao et al., 2017; Ke et al., 2018). Recently, discrete distributions have emerged as a promising alternative, offering several compelling advantages, including mitigating the notorious posterior collapse issue (Bowman et al., 2016), enabling enhanced controllability through latent variable manipulation (Bartolucci et al., 2022), and demonstrating remarkable interpretability by revealing correspondences between latent variables and categorical language features like dialogue acts (Zhao et al., 2018), entity states (Guan et al., 2023), and writing actions (Cornille et al., 2024). The discrete latent variable models typically rely on a multinomial distribution over a learnable codebook (Van Den Oord et al., 2017) or a predefined vocabulary (Zelikman et al., 2024) to represent the discrete latent space. Despite the extensive exploration of latent variable models, their applications to the alignment of LLMs remains largely unexplored. Our work represents a pioneering effort to leverage the expressive power of discrete latent variables for capturing multifaceted human preferences. While a concurrent study by Poddar et al. (2024) employs continuous latent variables to represent various personalized needs, we aim to model the intricate preferences obscured in the prompts through the more interpretable approach of discrete latent variables.

## 2.2 LEARNING FROM HUMAN FEEDBACK

Learning from human feedback has been a crucial paradigm in aligning LLMs. Various forms of feedback have been explored, including labels (Hastie et al., 2009), scalar ratings (Silver et al., 2021; Richemond et al., 2024), expert trajectories (Hussein et al., 2017), and pairwise rankings (Wirth et al., 2017; Rafailov et al., 2024), all of which can be viewed as carriers of underlying human preferences. Recently, reward modeling techniques, particularly those based on pairwise rankings, have emerged as a promising approach for providing scalable feedback, such as the Bradley-Terry model (Bradley & Terry, 1952). Such reward models can then be leveraged to align LLMs with human preference through reinforcement learning algorithms like PPO (Schulman et al., 2017). This has been applied to ensure safety (Dai et al., 2024), enhance helpfulness (Nakano et al., 2021), and promote honesty (Tian et al., 2024) in LLMs. However, the complex implementation, hyperparameter tuning, sample inefficiency, and computational overhead of PPO (Choshen et al., 2020) have motivated the exploration of simpler approaches, including rejection sampling (Touvron et al., 2023) that fine-tunes LLMs on responses with the highest reward among a number of samples, and direct preference optimization (DPO) (Rafailov et al., 2024) that directly optimizes LLMs from human preference data without an explicit reward model. Following DPO, various preference optimization objectives have been proposed, such as KTO (Ethayarajh et al., 2024), DRO (Richemond et al., 2024), SimPO (Meng et al., 2024), and GPO (Tang et al., 2024). Despite these advancements, a common limitation of existing methods is their assumption of a single, unified reward function, which may fail to capture the multifaceted nature of human preferences.

## 2.3 MULTI-OBJECITVE OPTIMIZATION

Multi-objective optimization for aligning LLMs has garnered significant attention, as it mitigates potential dichotomies between competing objectives (Bai et al., 2022) and caters to diverse user needs (Dong et al., 2023). Existing approaches to multi-objective alignment can be broadly categorized into three groups: **(1) Reward Model Combination**, which transforms multi-objective alignment into a single-objective optimization problem by linearly combining rewards from individual reward models (Wu et al., 2023) or via parameter interpolation (Rame et al., 2023). Then, they use standard RL approaches to maximize the scalar reward. **(2) Policy Model Combination**, which applies the spirit of linear combination to policy models, i.e., combining policy models learned from different reward models through token-wise probability interpolation (Jang et al., 2023). **(3) Combination-aware Learning**, which trains a single policy model conditioned on both the user instruction and the expected combination weights of different objectives (Dong et al., 2023; Wang et al., 2024). All these methods require explicit human feedback for each objective and demand prespecified weights for combining multi-objective rewards, imposing a substantial burden on human annotators. In contrast, our approach aims to automatically infer both the implicit factors and their relative importance from holistic feedback data, without relying on pre-defined objective weights or explicit reward models.

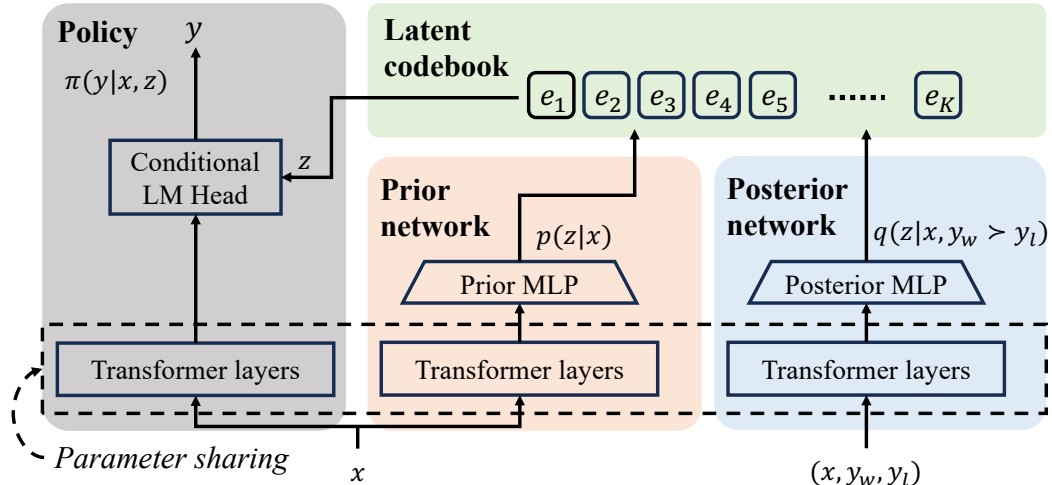

Figure 1: Overview of **Latent Preference Coding**. The framework is comprised of a discrete codebook and three modules: a policy model $\pi_\theta(y|x, z)$ conditioned on a latent variable $z$, a prior network $p(z|x)$ that learns to infer $z$ from the prompt, and a posterior network $q(z|x, y_w \succ y_l)$ that guides the training of the prior network and latent code embeddings.

## 3 METHODOLOGY

We elaborate Latent Preference Coding (LPC) in this section. Starting from a brief review of existing efforts in reinforcement learning from human feedback (RLHF) (§3.2), we derive the optimization objective of LPC (§3.2), and then formulate the latent representation of preferences and other important components in LPC (§3.3). Finally, we demonstrate how LPC can be seamlessly integrated into a variety of offline RLHF algorithms (§3.4).

### 3.1 PRELIMINARIES: REINFORCEMENT LEARNING FROM HUMAN FEEDBACK

The goal of RLHF is to optimize a language model $\pi_\theta(y|x)$ parameterized by $\theta$, initialized from a reference model $\pi_{\text{ref}}(y|x)$ obtained through pre-training or supervised fine-tuning. The optimization of $\pi_\theta(y|x)$ is guided by a reward model parameterized as $r_\phi(x, y)$, whose responsibility is to evaluate how well the output $y \sim \pi_\theta(y|x)$ aligns with human preference. Specifically, the policy model $\pi_\theta(y|x)$ is optimized to maximize the expected reward from $r_\phi(x, y)$ while constrained by a KL penalty with respect to the reference model $\pi_{\text{ref}}(y|x)$ (Ouyang et al., 2022):

$$\max_{\pi_\theta} \mathbb{E}_{x \sim \mathcal{D}, y \sim \pi_\theta(y|x)}[r_\phi(x, y)] - \beta \cdot \mathbb{D}_{\text{KL}}[\pi_\theta(y|x)||\pi_{\text{ref}}(y|x)], \quad (1)$$

where $\beta$ acts as a trade-off between the expectation of the reward and the KL term.

Normally, $r_\phi(x, y)$ is estimated from a preference dataset $\mathcal{D} = \{(x^i, y_w^i, y_l^i)\}_{i=1}^N$ by optimizing a Bradley-Terry (BT) model (Bradley & Terry, 1952):

$$\mathbb{E}_{(x, y_w, y_l) \sim \mathcal{D}} \log p(y_w \succ y_l|x) = \mathbb{E}_{(x, y_w, y_l) \sim \mathcal{D}} \log[\sigma(r_\phi(x, y_w) - r_\phi(x, y_l))], \quad (2)$$

where for prompt $x_i$, completion $y_w^i$ is more preferred than $y_l^i$.

Problem 1 often requires a complex and unstable online algorithm (Schulman et al., 2017), which motivates the exploration on offline RLHF. In fact, as pointed out in Go et al. (2023), the solution to the KL-constrained reward maximization objective 1 can be analytically written as:

$$\pi_\theta^\star(y|x) = \frac{1}{Z(x)} \pi_{\text{ref}}(y|x) \exp(\beta^{-1} r_\phi(x, y)), \quad (3)$$

where $Z(x)$ is the partition function. Hence, reward $r_\phi(x, y)$ can be represented by:

$$r_\phi(x, y) = \beta \log\left(\frac{\pi_\theta^\star(y|x)}{\pi_{\text{ref}}(y|x)}\right) + \beta \log(Z(x)). \quad (4)$$

Putting Eq. 2 and Eq. 4 together, RLHF can be performed offline without the need of an explicit reward by learning from the following loss (Rafailov et al., 2024):

$$\mathcal{L}_{\text{DPO}} = -\mathbb{E}_{(x,y_w,y_l)\sim\mathcal{D}} \log p(y_w \succ y_l|x)$$
$$= -\mathbb{E}_{(x,y_w,y_l)\sim\mathcal{D}} \left[ \log \sigma \left( \beta \log \frac{\pi_\theta(y_w|x)}{\pi_{\text{ref}}(y_w|x)} - \beta \log \frac{\pi_\theta(y_l|x)}{\pi_{\text{ref}}(y_l|x)} \right) \right]. \tag{5}$$

Due to its simplicity and effectiveness, offline RLHF has been adopted in the development of several leading LLMs (Touvron et al., 2023; Dubey et al., 2024; Yang et al., 2024a). Therefore, we choose offline RLHF as the starting point for our research on preference modeling, and leave the exploration for online RLHF as future work.

## 3.2 LEARNING OBJECTIVE OF LATENT PREFERENCE CODING

Recognizing the diverse and multifaceted nature of human preferences, our approach deviates from traditional RLHF methods that rely on a single reward model $r_\phi(x, y)$ to evaluate all data instances (either explicitly (Schulman et al., 2017) or implicitly (Rafailov et al., 2024)). Instead, we aim to capture the factors that underpin intricate holistic human preferences. To this end, two problems must be addressed: (1) *How to model the mixture of factors implied by a prompt?* And (2) *How to automatically and effectively learn the mixtures of factors from data in an unsupervised fashion?* To answer these questions, we propose latent preference coding (LPC) that implicitly models the underlying factors behind human preferences using latent variables.

We assume that holistic human preference is a mixture of multiple unobserved factors, and can be modeled by a latent variable $z$. Hence, the preference model $p(y_w \succ y_l|x)$ in Eq. 5 can be factorized as $p(y_w \succ y_l|z, x) \cdot p(z|x)$, where $p(z|x)$ is a prior modeling the induction of a mixture of factors as a specific preference pattern with respect to prompt $x$, and $p(y_w \succ y_l|z, x)$ measures how $y_w$ is preferred over $y_l$ under the prompt and the preference pattern. Following the assumption, the loss given by Eq. 5 can be re-formulated as:

$$\mathcal{L}_{\text{LPC-DPO}} = -\mathbb{E}_{(x,y_w,y_l)\sim\mathcal{D}} \log p(y_w \succ y_l|x)$$
$$= -\mathbb{E}_{(x,y_w,y_l)\sim\mathcal{D}} \log \mathbb{E}_{z\sim p(z|x)} p(y_w \succ y_l|x, z)$$
$$= -\mathbb{E}_{(x,y_w,y_l)\sim\mathcal{D}} \left[ \log \mathbb{E}_{z\sim p(z|x)} \sigma \left( \beta \log \frac{\pi_\theta(y_w|x, z)}{\pi_{\text{ref}}(y_w|x, z)} - \beta \log \frac{\pi_\theta(y_l|x, z)}{\pi_{\text{ref}}(y_l|x, z)} \right) \right], \tag{6}$$

where $\pi_{\{\theta,\text{ref}\}}(y|x, z)$ represent policy models conditioned on $z$ that will be detailed later.

Normally, it is difficult to directly optimize Eq. 6 due to the intractability of $p(z|x)$. Therefore, we consider a posterior $q(z|x, y_w \succ y_l)$ and perform learning through variational inference. The posterior takes the observed preference between $y_w$ and $y_l$ as input and predicts a distribution of $z$, which is then used to guide the direction of the prior. By this means, the negative evidence lower bound (ELBO) for $\mathcal{L}_{\text{LPC-DPO}}$ is given by:

$$\tilde{\mathcal{L}}_{\text{LPC-DPO}} = -\mathbb{E}_{(x,y_w,y_l)\sim\mathcal{D}} \left[ \mathbb{E}_{z\sim q(\cdot|x,y_w\succ y_l)} \log \sigma \left( \beta \log \frac{\pi_\theta(y_w|x, z)}{\pi_{\text{ref}}(y_w|x)} - \beta \log \frac{\pi_\theta(y_l|x, z)}{\pi_{\text{ref}}(y_l|x)} \right) \right.$$
$$\left. - \lambda \mathbb{D}_{\text{KL}}[q(\cdot|x, y_w \succ y_l)||p_z(\cdot|x)] \right], \tag{7}$$

where $\lambda$ is a hyper-parameter. Details of derivation are presented in Appendix A.

During inference, LPC first samples a latent variable $z$ according to the prior $p(z|x)$, and then generates the completion $y$ from $\pi_\theta(y|x, z)$.

## 3.3 MODELING OF LATENT PREFERENCE CODING

Figure 1 illustrates the architecture of LPC. To effectively represent the multifaceted factors that shape holistic human preferences, we propose to model the underlying factors through discrete latent variables. Unlike continuous representations, discrete latent variables have been shown to mitigate the notorious posterior collapse issue (Bowman et al., 2016) and enjoy better interpretability (Guan et al., 2023). Basically, we implement LPC based on the standard decoder-only Transformer

architecture (Vaswani et al., 2017) parameterized by $\theta$, taking the input prompt $x$ and generating the output completion $y$. We use $\boldsymbol{h}_x$ and $\boldsymbol{h}_{x,y}$ to denote the hidden states of the last layer at the last token of $x$ and the concatenation of $x$ and $y$, respectively.

**Discrete Latent Space.** We introduce a discrete codebook $E = \{\boldsymbol{e}_k \in \mathbb{R}^d\}_{k=1}^K$ that comprising $K$ codes, where each code $\boldsymbol{e}_k$ corresponds to an underlying factor influencing the holistic preference. We assume that both the prior and posterior distributions are categorical distributions over the latent codes in $E$, making it easy to derive the KL divergence between them in Eq. 7.

**Posterior network.** Given a triple of $(x, y_w, y_l)$, we implement the posterior network by applying a two-layer MLP on the concatenation of $\boldsymbol{h}_{x,y_w}$ and $\boldsymbol{h}_{x,y_l}$:

$$q(z|x, y_w \succ y_l) = \text{softmax}\left(\text{MLP}_{\text{posterior}}([\boldsymbol{h}_{x,y_w}; \boldsymbol{h}_{x,y_l}])\right). \tag{8}$$

**Prior network.** Given an input prompt $x$, the prior network feeds $\boldsymbol{h}_x$ to another MLP, which predicts the prior distribution over the latent codes:

$$p(z|x) = \text{softmax}\left(\text{MLP}_{\text{prior}}(\boldsymbol{h}_x)\right). \tag{9}$$

**Policy Model.** To effectively leverage the insights gained from LPC, the policy model should seamlessly integrate the holistic preference representation derived from the latent variable $z$ into the language generation process. Formally, we model the conditional probability $\pi_\theta(y|x, z)$ as follows:

$$\pi_\theta(y|x, z) = \prod_t \pi_\theta(y_t, |x, z, y_{<t})$$

$$= \prod_t \text{softmax}(\text{LMHead}(\boldsymbol{h}_{x,y_{<t}} + \boldsymbol{z})), \tag{10}$$

where LMHead is the language model head mapping the hidden states to the vocabulary, $\boldsymbol{h}_{x,y_{<t}}$ denotes the hidden state of the language model encoding the prompt $x$ and the partially generated completion $y_{<t}$, and $\boldsymbol{z}$ denotes the representation of the holistic human preference derived from the prior or posterior distributions of the latent variable $z$.

To circumvent the non-differentiability of sampling from discrete categorical distributions, we leverage the Gumbel-softmax reparameterization trick (Jang et al., 2017), which allows us to obtain continuous and differentiable samples from the prior and posterior distributions over the latent codes. Specifically, we derive $\boldsymbol{z}$ as a convex combination of all latent code embeddings in $E$, weighted by the Gumbel-softmax samples from the prior and the posterior distributions:

$$\boldsymbol{z} = \sum_{k=1}^K c_k \boldsymbol{e}_k,$$

$$\{c_k\}_{k=1}^K = g \cdot \text{Gumbel-softmax}(p(z|x))$$
$$+ (1 - g) \cdot \text{Gumbel-softmax}(q(z|x, y_w \succ y_l)), \tag{11}$$

where $\{c_k\}_{k=1}^K$ is the categorical distribution over the latent codes after applying Gumbel-softmax on the prior or posterior distributions, and $g \in [0, 1]$ is a weight that determines the relative contributions of the prior and the posterior distributions in deriving $\boldsymbol{z}$. We employ a linear scheduling strategy to gradually increase $g$ from 0 to 1 during training, allowing the model to initially rely more on the more accurate posterior distribution for guidance, and progressively shift towards the prior distribution as the training goes on. In this way, LPC can automatically infer their relative importance between different underlying factors during training.

### 3.4 EXTENSION TO OTHER OFFLINE RLHF OBJECTIVES

While the derivation of LPC originates from the DPO objective, its versatile formulation readily extends to other offline RLHF objectives if the obejctives can be formulated as $-\log(f(\cdot))$. This enables a unified framework for capturing the intricate nature of human preferences across different optimization paradigms.

Specifically, when applying LPC to SimPO (Meng et al., 2024), we derive the following loss:

$$\mathcal{L}_{\text{LPC -SimPO}} = -\mathbb{E}_{(x,y_w,y_l)\sim\mathcal{D}}\Bigg[\mathbb{E}_{z\sim q(\cdot|x,y_w\succ y_l)}\log\sigma\Bigg(\frac{\beta}{|y_w|}\log\pi_\theta(y_w|x,z)$$
$$-\frac{\beta}{|y_l|}\log\pi_\theta(y_l|x,z)-\gamma\Bigg)-\lambda\mathbb{D}_{\text{KL}}[q(\cdot|x,y_w\succ y_l)||p_z(\cdot|x)]\Bigg]. \quad (12)$$

Furthermore, drawing inspiration from Eq. 7, we can also apply LPC to objectives that do not strictly satisfy $-\log(f(\cdot))^2$. While the extension sacrifices some mathematical rigor, it proves beneficial in practice, as will be seen in Experiments. Specifically, when applied to IPO (Azar et al., 2024), the loss for learning is given by:

$$\mathcal{L}_{\text{LPC -IPO}} = \mathbb{E}_{(x,y_w,y_l)\sim\mathcal{D}}\Bigg[\mathbb{E}_{z\sim q(\cdot|x,y_w\succ y_l)}\Bigg(\log\frac{\pi_\theta(y_w|x,z)}{\pi_{\text{ref}}(y_w|x)}-\log\frac{\pi_\theta(y_l|x,z)}{\pi_{\text{ref}}(y_l|x)}-\frac{1}{2\tau}\Bigg)^2$$
$$+\lambda\mathbb{D}_{\text{KL}}[q(\cdot|x,y_w\succ y_l)||p_z(\cdot|x)]\Bigg]. \quad (13)$$

Similarly, one can also extends LPC to more objectives as presented in Tang et al. (2024).

## 4 EXPERIMENTS

### 4.1 EXPERIMENTAL SETUP

**Configuration.** To comprehensively evaluate the efficacy of LPC, we conduct experiments using three open-source LLMs: Mistral-7B (Jiang et al., 2023), Llama3-8B, and Llama3-8B-Instruct (Dubey et al., 2024). Furthermore, to demonstrate the compatibility and flexibility of LPC, we integrate it with three widely used offline preference learning algorithms: DPO, IPO, and SimPO. These algorithms encompass different optimization strategies and inductive biases, enabling a comprehensive evaluation of LPC's performance across diverse preference learning methods. In total, we investigate nine distinct configurations, resulting from the combination of three base models and three preference learning algorithms. For each configuration, we compare against two baselines: the base model and its optimized version using the corresponding preference learning algorithm.

**Dataset.** We utilize the widely-adopted UltraFeedback dataset (Cui et al., 2023) in experiments. The dataset is a comprehensive collection of user preferences spanning diverse domains. It contains 63,967 instances from 6 publicly available datasets, including TruthfulQA, FalseQA, Evol-Instruct, UltraChat, ShareGPT, and FLAN. We randomly sample 1,000 instances for validation and an additional 1,000 instances for testing. The rest of the instances are used for training LPC and the baseline alignment methods. We adopt the same data preprocessing pipeline as outlined in (Tunstall et al., 2023) to construct the preference pairs. For each instance, four completions are generated by different LMs. The completion with the highest overall score is denoted as $y_w$, while $y_l$ is randomly sampled from the remaining completions.

**Evaluation.** We first evaluate LPC and the baselines on several representative downstream benchmarks in terms of three aspects: (1) Commonsense Reasoning: we employ ARC-challenge and ARC-easy (Clark et al., 2018) as the evaluation datasets. (2) Mathematical Reasoning: GSM8K (Cobbe et al., 2021), a collection of grade-school problems, is exploited for evaluation. (3) Truthfulness: we use TruthfulQA (Lin et al., 2022) to assess the honesty of aligned LLMs.

Then, we assess how well the models capture the holistic human preferences by calculating the preference accuracy for ranking completion pairs. Specifically, the accuracy accounts for the proportion of instances where $y_w$ has a higher reward score than $y_l$ based on Eq. 4. We calculate the preference accuracy on the test set of UltraFeedback comprising 1,000 examples.

**Implementation Details.** We leverage the OpenRLHF library (Hu et al., 2024) for model training. All models are trained for one epoch, employing the AdamW optimizer (Loshchilov, 2017) and a

---

[2]In this case, a rigorous derivation of the KL term is not feasible. Therefore, we retain the KL term in Eq. 7 and just replace the expectation term analogously to the formulation used in LPC for DPO.

Table 1: Evaluation results on the downstream tasks.

| Method | Arc-challenge (0-shot) | Arc-easy (0-shot) | GSM8K (5-shot) | TruthfulQA (0-shot) | Average |
|---|---|---|---|---|---|
| **Mistral-7B** | | | | | |
| *Base Model* | *49.74* | *80.72* | *37.30* | *41.13* | *52.22* |
| **DPO** | 55.38 | 83.33 | 40.11 | **48.10** | 56.73 |
| **DPO w. LPC** | **55.55** | **83.54** | **44.28** | 47.86 | **57.81** (+1.08) |
| **IPO** | **58.19** | **84.76** | 30.48 | 48.96 | 55.60 |
| **IPO w. LPC** | 57.17 | 83.88 | **42.61** | **50.80** | **58.61** (+3.01) |
| **SimPO** | **58.11** | **84.68** | 31.01 | 49.33 | 55.78 |
| **SimPO w. LPC** | 56.40 | 83.59 | **32.60** | **51.29** | **55.97** (+0.19) |
| **Llama3-8B** | | | | | |
| *Base Model* | *50.43* | *80.05* | *49.51* | *43.82* | *55.95* |
| **DPO** | 54.01 | 81.27 | 54.36 | 43.70 | 58.33 |
| **DPO w. LPC** | **54.18** | **81.48** | **55.34** | **44.68** | **58.92** (+0.59) |
| **IPO** | 51.37 | **80.89** | 50.42 | 44.19 | 56.72 |
| **IPO w. LPC** | **51.54** | 80.81 | **50.95** | **45.41** | **57.18** (+0.46) |
| **SimPO** | **54.95** | **81.90** | **46.02** | 39.53 | 55.60 |
| **SimPO w. LPC** | 53.33 | 81.36 | 45.87 | **53.61** | **58.54** (+2.94) |
| **Llama3-8B-Instruct** | | | | | |
| *Base Model* | *52.90* | *81.52* | *75.66* | *46.88* | *64.24* |
| **DPO** | 54.35 | 82.24 | 77.03 | 47.00 | 65.16 |
| **DPO w. LPC** | **55.29** | **82.41** | **77.79** | **48.10** | **65.90** (+0.74) |
| **IPO** | 53.84 | 81.57 | 73.69 | **47.25** | 64.09 |
| **IPO w. LPC** | **54.69** | **82.24** | **76.80** | 46.51 | **65.06** (+0.97) |
| **SimPO** | 55.97 | **83.50** | 66.79 | **56.79** | 65.77 |
| **SimPO w. LPC** | **57.34** | 82.91 | **73.62** | 56.06 | **67.48** (+1.71) |

linear learning rate scheduler peaking at 5e-7 with a 10% warm-up phase. The global batch size is set to 64 and the max length is 1,024. For LPC, we search $\lambda$ in Eq.7 from $\{0.01, 0.05, 0.1\}$ and find $\lambda = 0.05$ yields good performance across all methods. For the DPO and SimPO methods, we regulate the deviation from the reference model by setting $\beta$ in Eq. 5 and Eq. 12 to 0.1. In the case of IPO, we explore the optimal $\tau$ value in Eq. 13 from $\{0.01, 0.05, 0.1, 0.5\}$ based on the validation performance and empirically choose $\tau = 0.01$. For downstream task evaluation, we utilize the Language Model Evaluation Harness library (Gao et al., 2024), adhering to the default hyper-parameters and evaluation settings.

## 4.2 MAIN RESULTS

**Downstream Benchmark Evaluation.**    As demonstrated in Table 1, it is evident that the proposed LPC framework consistently enhances the performance of LLMs across a diverse range of downstream tasks, base models, and preference methods. A closer examination of the results reveals several key insights: (1) DPO emerges as the most robust alignment method, yielding consistent performance gain over the base models across all datasets. Notably, when augmented with LPC, DPO's performance is further amplified, accentuating the synergistic benefits of LPC in modeling the underlying preference factors. (2) SimPO and IPO exhibit more variability in their performance, occasionally underperforming the base models on certain tasks, particularly GSM8K. However, when integrated with LPC, these performance deficits are mitigated, and in some cases, even surpassed (e.g., IPO w. LPC for Mistral-7B and Llama3-8B-Instruct on GSM8K, and SimPO w. LPC for Llama3-8B on TruthfulQA), underscoring LPC's ability to elucidate and harmonize the disparate preference factors. (3) LPC's impact is not uniformly distributed across all tasks. Specifically, on

Table 2: Comparison of preference accuracy before and after integrating LPC.

| Algorithm | Llama3-8B | Llama3-8B-Instrcut | Mistral-7B |
|---|---|---|---|
| **DPO** | 69.3 / **70.8** (+1.5) | **70.1** / 69.9 (-0.2) | 71.9 / **73.4** (+1.5) |
| **IPO** | 68.0 / **70.6** (+2.6) | 68.3 / **70.3** (+2.0) | 74.2 / **74.7** (+0.5) |
| **SimPO** | 69.2 / **71.8** (+2.6) | **74.1** / 73.2 (-0.9) | 73.5 / **75.6** (+2.1) |

tasks that heavily rely on the model's intrinsic capabilities, such as abstraction and reasoning skills in the case of the ARC datasets, LPC's fine-grained preference modeling yields relatively modest improvements. This suggests that while LPC excels in capturing the nuances of human preferences, it may have a limited influence on enhancing the model's commonsense reasoning capabilities, which are primarily shaped during the pre-training stage.

**Preference Accuracy.** Subsequently, we delve into the preference accuracy evaluation to assess the efficacy of LPC in distinguishing between favorable and unfavorable completions. As presented in Table 2, the integration of LPC generally elevates preference accuracy across various base models and alignment algorithms. This empirical evidence corroborates LPC's capacity to elucidate and harmonize the intricate factors that shape human preferences. Notably, for the Llama3-8B-Instruct model, the impact of LPC on preference accuracy appears relatively muted. We conjecture that this is because Llama3-8B-Instruct has been extensively fine-tuned for instruction-following, which imbues it with an enhanced ability to adhere to diverse human instructions. Consequently, the influence of LPC's fine-grained preference modeling may be somewhat constrained. Nevertheless, as aforementioned, LPC continues to confer substantial performance improvements on downstream tasks, even for Llama3-8B-Instruct, underscoring its versatility and robustness.

### 4.3 Latent Code Analysis

To gain deeper insights into the proposed LPC framework, we conduct experiments to unravel two pivotal research questions: (1) What is the optimal size of the latent codebook to effectively capture the intricate landscape of human preferences? (2) Does LPC truly capture the implicit factors underpinning holistic preferences as hypothesized?

**Investigating the Optimal Codebook Size.** To investigate the optimal codebook size, we train a series of models with distinct codebook sizes ranging from the set $\{8, 16, 32, 64, 128, 256\}$. As illustrated in Figure 2 (Top Left), the preference accuracy exhibits a distinct pattern: initially increasing with larger codebook sizes, peaking around 32 to 64 codes, and then gradually declining, suggesting that it is crucial for LPC's performance to striking the right balance in the size of the latent codebook. When the codebook is small (e.g., 8 codes), it may be insufficiently expressive to capture the diversity of implicit preference factors, thereby limiting performance. Conversely, when the codebook is excessively large (e.g., 256 codes), LPC does not appear to derive significant benefits from an expanded latent space. This could be attributed to several factors: (1) the model may struggle to effectively utilize such a high-dimensional latent space given the limited training data, or (2) the risk of overfitting increases as the codebook size grows.

**Investigating the Capability to Distinguish Implicit Factors.** The core rationale behind predicting implicit preference factors using the prior network $p(z|x)$ lies in the assumption that the prompt $x$ accurately reflects the underlying preference structure. To validate this critical assumption, we devise a probing experiment by intentionally distorting the preference annotations in the UltraFeedback dataset. Specifically, we randomly flip 50% of the preference labels in the training data (i.e., replacing "$y_w \succ y_l$" with "$y_w \prec y_l$"), appending a special token [FLIP] to the prompts associated with these flipped instances. Subsequently, we train Llama3-8B using DPO with or without LPC on this distorted dataset to assess whether LPC can effectively differentiate between flipped preferences and normal ones. Then, we calculate the preference accuracy on the original test set of UltraFeedback. As illustrated in Figure 2 (Bottom Left), LPC outperforms the baseline DPO by a larger margin than in the ordinary setting, indicating that in more complex preference environments with intermixed preferences—some of which are even completely opposite—LPC's capability to

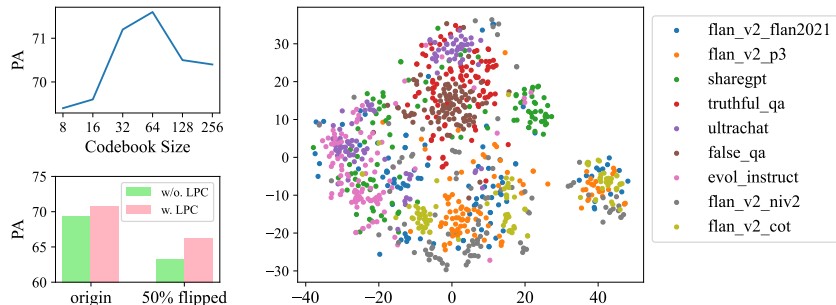

Figure 2: **Top left:** Preference accuracy (PA) of DPO w. LPC on Llama3-8B varying with the latent codebook size. **Bottom Left:** Flipping-label experiment on Llama3-8B. Models are evaluated on the original test set with unflipped labels. **Right:** Visualization of the latent variable $z$ produced by the prior network of Llama3-8B. The alignment method is DPO. For each data source in UltraFeedback, we randomly select 100 instances and visualize the T-SNE features of these instances.

Table 3: Results on AlpacaEval 2 judged by GPT-4-turbo-2024-04-09. "LC" and "WR" denote length-controlled and raw win rate, respectively. All methods are based on Llama3-8B-Instruct. The baseline model compared against is GPT-4-1106-preview. We use the official evaluation script (Li et al., 2023), adopting the same decoding hyper-parameters as Meng et al. (2024), with the temperature set to 0.9.

| Algorithm | DPO | | IPO | | SimPO | |
|---|---|---|---|---|---|---|
| | LC(%) | WR(%) | LC(%) | WR(%) | LC(%) | WR(%) |
| **w/o. LPC** | 15.03 | 13.55 | 14.44 | 12.96 | 12.77 | 8.12 |
| **w. LPC** | **15.31** | **13.57** | **15.04** | **13.50** | **15.56** | **9.63** |

model implicit preference factors enables it to distinguish and disentangle these conflicting signals, thereby enhancing overall performance.

Additionally, we employ T-SNE (Van der Maaten & Hinton, 2008) to visualize the latent variable $z$. As depicted in Figure 2 (Right), instances from different data sources cluster into several distinct groups. This clustering phenomenon arises because data from various sources typically emphasize different preferences. This observation further corroborates the effectiveness of LPC in modeling implicit preference factors, as it can capture the intricate preference structures inherent in diverse data sources.

### 4.4 WIN RATE AGAINST GPT-4

To further validate LPC's efficacy in aligning LLMs with human preferences, we evaluate LPC on AlpacaEval 2 (Li et al., 2023) using GPT-4 as a judge. As depicted in Table 3, LPC brings performance improvements across all alignment algorithms on Llama3-8B-Instruct, further solidifying its prowess in aligning LLMs with human preferences.

### 5 CONCLUSIONS

In this work, we propose LPC, a framework that enables LLMs to capture the multifaceted nature of human preferences. LPC introduces discrete latent codes where each code represents an underlying factor influencing holistic preferences. Through variational inference, LPC can model the implicit factors without the need for fine-grained preference annotations. Besides, LPC can be integrated with a variety of offline preference algorithms, including DPO, IPO, SimPO, and so on. We conduct extensive experiments evaluating LPC on three open-source LLMs, showing that LLMs can achieve better performance across multiple benchmarks by modeling the underlying factors of human preference.

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

## A  DERIVING THE EVIDENCE LOWER BOUND OF LPC

We start with the standard DPO, where the objective is to maximize the log-likelihood $\mathbb{E}_{(x,y_w,y_l)\sim\mathcal{D}} \log p(y_w \succ y_l|x)$. After introducing latent variable $z$, we have:

$$
\begin{aligned}
\log p(y_w \succ y_l|x) &= \log \mathbb{E}_{z\sim p(z|x)} p(y_w \succ y_l|x,z) \\
&= \log \int p(y_w \succ y_l|x,z) p(z|x) \frac{q(z|x, y_w \succ y_l)}{q(z|x, y_w \succ y_l)} dz \\
&= \log \mathbb{E}_{z\sim q(\cdot|x,y_w\succ y_l)} \frac{p(y_w \succ y_l|x,z) p(z|x)}{q(z|x, y_w \succ y_l)} \\
&\geq \mathbb{E}_{z\sim q(\cdot|x,y_w\succ y_l)} \log \frac{p(y_w \succ y_l|x,z) p(z|x)}{q(z|x, y_w \succ y_l)} \\
&= \mathbb{E}_{z\sim q(\cdot|x,y_w\succ y_l)} \left[ \log p(y_w \succ y_l|x,z) + \log \frac{p(z|x)}{q(z|x, y_w \succ y_l)} \right] \\
&= \mathbb{E}_{z\sim q(\cdot|x,y_w\succ y_l)} \log p(y_w \succ y_l|x,z) - \mathbb{D}_{\text{KL}}[q(\cdot|x, y_w \succ y_l)||p(\cdot|x)].
\end{aligned}
\tag{14}
$$

By this means, we have:

$$
\tilde{\mathcal{L}}_{\text{LPC-DPO}} = -\mathbb{E}_{(x,y_w,y_l)\sim\mathcal{D}} \left[ \mathbb{E}_{z\sim q(\cdot|x,y_w\succ y_l)} \log p(y_w \succ y_l|x,z) - \mathbb{D}_{\text{KL}}[q(\cdot|x, y_w \succ y_l)||p(\cdot|x)] \right].
\tag{15}
$$

Then we need to derive the mathematical solution for $p(y_w \succ y_l|x,z)$. We assume that each latent $z$ corresponds to an implicit reward model $r_{\phi_z}(x,y)$. The derivation process is quite similar to standard DPO.

For each implicit preference factor $z$, we optimize the following objective:

$$
\max_{\pi_\theta} \mathbb{E}_{x\sim\mathcal{D}, y\sim\pi_\theta(\cdot|x,z)}[r_{\phi_z}(x,y)] - \beta \mathbb{D}_{\text{KL}}[\pi_\theta(y|x,z)||\pi_{\text{ref}}(y|x,z)].
\tag{16}
$$

Because the parameter of the reference model is fixed during training, the output of $\pi_{\text{ref}}$ would not be affected by $z$, i.e., $\pi_{\text{ref}}(y|x,z) = \pi_{\text{ref}}(y|x)$. We now have:

$$
\begin{aligned}
&\max_{\pi_\theta} \mathbb{E}_{x\sim\mathcal{D}, y\sim\pi_\theta(\cdot|x,z)}[r_{\phi_z}(x,y)] - \beta \mathbb{D}_{\text{KL}}[\pi_\theta(y|x,z)||\pi_{\text{ref}}(y|x,z)] \\
&= \max_{\pi_\theta} \mathbb{E}_{x\sim\mathcal{D}, y\sim\pi_{\theta_z}(\cdot|x,z)} \left[ r_{\phi_z}(x,y) - \beta \log \frac{\pi_{\theta_z}(y|x,z)}{\pi_{\text{ref}}(y|x)} \right] \\
&= \min_{\pi_{\theta_z}} \mathbb{E}_{x\sim\mathcal{D}, y\sim\pi_{\theta_z}(\cdot|x,z)} \left[ \log \frac{\pi_{\theta_z}(y|x,z)}{\pi_{\text{ref}}(y|x)} - \beta^{-1} r_{\phi_z}(x,y) \right] \\
&= \min_{\pi_{\theta_z}} \mathbb{E}_{x\sim\mathcal{D}, y\sim\pi_{\theta_z}(\cdot|x,z)} \left[ \log \frac{\pi_{\theta_z}(y|x,z)}{\pi^*(y|x,z)} - \log Z_z(x) \right] \\
&= \min_{\pi_{\theta_z}} \mathbb{E}_{x\sim\mathcal{D}} [\mathbb{D}_{\text{KL}}(\pi_{\theta_z}(y|x,z)||\pi^*(y|x,z)) - \log Z_z(x)],
\end{aligned}
\tag{17}
$$

where:

$$
Z_z(x) = \sum_y \pi_{\text{ref}}(y|x) \exp\left(\beta^{-1} r_{\phi_z}(x,y)\right)
\tag{18}
$$

and

$$
\pi^*(y|x,z) = \frac{1}{Z_z(x)} \pi_{\text{ref}}(y|x) \exp\left(\beta^{-1} r_{\phi_z}(x,y)\right).
\tag{19}
$$

The KL-divergence in Eq.17 reaches the minimum when $\pi_{\theta_z}(y|x,z) = \pi^*(y|x,z))$. As a result, we obtain the expression of optimal reward:

$$
r^*_{\phi_z}(x,y) = \beta \log \frac{\pi^*(y|x,z)}{\pi_{\text{ref}}(y|x)} + \beta \log Z_z(x).
\tag{20}
$$

Combining Eq.15 and Eq.20, we can get the final training objective of LPC.

$$
\begin{aligned}
\tilde{\mathcal{L}}_{\text{LPC-DPO}} &= -\mathbb{E}_{(x,y_w,y_l)\sim\mathcal{D}}\left[\mathbb{E}_{z\sim q(\cdot|x,y_w\succ y_l)}\log p(y_w\succ y_l|x,z) - \mathbb{D}_{\text{KL}}[q(\cdot|x,y_w\succ y_l)||p(\cdot|x)]\right], \\
&= -\mathbb{E}_{(x,y_w,y_l)\sim\mathcal{D}}\left[\mathbb{E}_{z\sim q(\cdot|x,y_w\succ y_l)}\log\sigma(r_{\phi_z}(x,y_w) - r_{\phi_z}(x,y_l)) - \mathbb{D}_{\text{KL}}[q(\cdot|x,y_w\succ y_l)||p(\cdot|x)]\right], \\
&= -\mathbb{E}_{(x,y_w,y_l)\sim\mathcal{D}}\left[\mathbb{E}_{z\sim q(\cdot|x,y_w\succ y_l)}\log\sigma\left(\beta\log\frac{\pi_\theta(y_w|x,z)}{\pi_{\text{ref}}(y_w|x)} - \beta\log\frac{\pi_\theta(y_l|x,z)}{\pi_{\text{ref}}(y_l|x)}\right)\right. \\
&\qquad\qquad\left. - \mathbb{D}_{\text{KL}}[q(\cdot|x,y_w\succ y_l)||p(\cdot|x)]\right],
\end{aligned}
\tag{21}
$$

In practice, we insert a hyper-parameter $\lambda$ before the KL term to enhance flexibility of learning.