# OpenReview forum: "Latent Preference Coding: Aligning Large Language Models via Discrete Latent Codes"
_ICLR.cc/2025/Conference — Submitted to ICLR 2025_

### Official Review · Reviewer_YaiJ · 2024-10-23

**Soundness:** 3
**Presentation:** 2
**Contribution:** 3
**Rating:** 6
**Confidence:** 3

**Summary:**

This paper proposes LPC for solving the problem of aligning large language models with human preferences. LPC captures the multifaceted nature of human preferences through the introduced discrete latent variables. In addition, LPC can be integrated with a variety of offline preference algorithms, including DPO, IPO, SimPO, and so on. A unified approach is provided to represent complex preference structures, which in turn enhances the performance of LLMs.

**Strengths:**

1. The overall framework of the paper is relatively novel, capturing the multifaceted nature of human preferences with the help of discrete latent variables.
2. The constructed LPC framework can seamlessly integrate multiple existing alignment methods with high applicability and scalability.

**Weaknesses:**

1. The discussion and analysis of existing work is relatively brief.

2. Generalization ability is unclear: While LPC performs well on noisy data, the validation of its ability to generalize across tasks is more underdeveloped and does not demonstrate its adaptability to a broader range of tasks.

3. Insufficient discussion of LPC complexity: LPC methods' computational cost and implementation complexity in practical applications are not discussed in detail, especially the computational resource requirements for large-scale models.

**Questions:**

1. How do human preferences and discrete potential coding correspond in the LPC framework? How do we distinguish different preference elements?

2. What is the specific task division between a priori and a posteriori network, and what is the purpose of designing them? How do they operate when there are no preferences? How are the cues designed?

3. What is the performance of LPC in dealing with extreme cases (e.g., extremely noisy data or cases where preference labels are entirely missing)?

---

> ### Author Response · Authors · 2024-11-19
> **Response to Reviewer YaiJ**
>
> Thanks for your detailed review.
>
> ### Regarding more discussion and analysis of existing works (Weakness 1)
> Thanks for your suggestion. We will add more discussions and analysis of existing works in the revised version. If you have any specific suggestions, please let us know.
>
> ### Regarding the generalization ability of LPC (Weakness 2)
>
> Thanks for raising this point. Table 1 presents the results of LPC on different tasks, showing that LPC performs well on a diverse set of tasks. Additionally, we conduct evaluations on a broader range of tasks in MMLU. MMLU includes a diverse set of tasks including various domains, which is a good indicator of the generalization ability of a model. While LPC does not significantly improve the performance of alignment algorithms on MMLU, it also does not hurt the performance, which is consistent with previous works [1]. We believe that this is due to the large domain gap between training and evaluation. The results are as follows (with DPO as the alignment algorithm):
>
> | Method      | MMLU | MT-bench |
> | ----------- | ----------- | ----------- |
> | Llama3-8B        |  62.10      |   -     |
> | Llama3-8B w/o. LPC         | 62.68       | 6.32       |
> | Llama3-8B w. LPC         | 62.54      |  6.36      |
> | Llama3-Instruct-8B         |  63.89      |    -    |
> | Llama3-Instruct-8B w/o. LPC         | 63.60      | 7.67       |
> | Llama3-Instruct-8B w. LPC         | 63.72      |  7.82     |
>
>
> [1] Meng Y, Xia M, Chen D. Simpo: Simple preference optimization with a reference-free reward[J]. arXiv preprint arXiv:2405.14734, 2024.
>
> ### Regarding the complexity of LPC (Weakness 3)
> The computational cost of LPC is little higher than DPO. In our architecture design, the policy model, the prior network, and the posterior network share the same backbone LM. For the input triple $<x,y_w,y_l>$, we only need to forward the backbone LM twice (once for $concat(x,y_w)$ and once for $concat(x,y_l)$), which is the same as DPO. We will provide detailed implementation details regarding the time complexity in the revised version.
> As for the computational resource requirements, We train all the models with 8 A100 GPUs.
>
> ### Regarding question 1
> Thanks for your question. LPC uses different latent codes to represent different human preference directions. As seen in the T-SNE visualization (Figure 2), instances from different data sources cluster in different regions, showing that preferences are distinguishable in the latent space.
>
>
> ### Regarding issues about the prior and the posterior network (Question 2)
> Let us explain the purpose of the prior network and the posterior network. Given the input $x$, we need the prior network to estimate a variable $z$ to generate the output $y$. The problem is how we train the prior network. We need a ground truth $z$ to supervise the training of the prior network. Thus, we leverage a posterior network whose target is to give a good guess of $z$ based on preference data $<x,y_w,y_l>$. Then, by minimizing the difference in outputs of the prior network and the posterior network, we can inject the preference signal into the prior network.
> > How do they operate when there are no preferences?
>
> When there are no preferences, preference learning is infeasible.
> We hope this explanation can answer your question. If you have further doubts, please feel free to let us know.
>
>
> ### Regarding LPC in extreme cases (Question 3)
> Thanks for this question. In scenarios with **extremely noisy data**, we construct noisy data by flipping some of the preference labels in the training data. The results in a scenario where two totally opposite preferences are mixed together. LPC achieves good performance in this case, outperforming DPO by a large margin, showing its robustness on noisy data.
>
> |  Flip ratio      | 10% | 50% | 90% |
> | ----------- | ----------- | ----------- | ----------- |
> | Llama3-8B w/o. LPC         | 68.9      | 63.2       | 66.3      |
> | Llama3-8B w. LPC         | 70.1      | 66.2      | 67.4     |
> | diff         | +1.2      | +3.0       | +1.1  |
>
>
> As for scenarios with **entirely missing labels**, it is impractical to perform preference optimization without preference labels.

---

> > ### Comment · Reviewer_YaiJ · 2024-11-26
> >
> > Thank you for your reply. You've answered all my questions. But I'll keep my score.

---

> > > ### Author Response · Authors · 2024-12-02
> > > **Thank you!**
> > >
> > > Dear Reviewer YaiJ,
> > >
> > > Thanks for your response.
> > >
> > > Authors of Submission 3825

---

### Official Review · Reviewer_wojJ · 2024-11-01

**Soundness:** 2
**Presentation:** 2
**Contribution:** 3
**Rating:** 6
**Confidence:** 3

**Summary:**

Latent Preference Coding presents an innovative approach that leverages discrete latent variables to model the underlying factors behind human preferences. By implicitly capturing these factors without relying on predefined objective weights or manually defined reward functions, LPC provides a unified framework compatible with various alignment algorithms.

**Strengths:**

1. The idea behind LPC is interesting. The framework eliminates the need for manually defined reward functions and predefined combination weights, a limitation present in many other methods.
2. Experiments were conducted using three different base models and three alignment algorithms, showcasing LPC’s broad applicability.

**Weaknesses:**

1. The training settings for the discrete latent space are not clearly specified.
2. The TSNE visualization lacks distinct clusters, making it difficult to discern what the latent code has actually learned.
3. Table 1 does not include results for MMLU, which is a benchmark on the OpenLLM leaderboard.
4. The scores reported in Table 3 are generally low. For example, fine-tuning on Llama3-8B-Instruct using SimPO often yields scores of 20+ as reported in SimPO paper. Could there be differences in settings affecting these results?
5. Additional benchmarks, such as MT-Bench or Arena-Hard, would enhance the comprehensiveness of the evaluations.

**Questions:**

1. In Equation (10), is \( h_{x,y} \) at each timestep summed with \( z \)?
2. The purpose of the flipped-label experiment is unclear. Could you clarify how LPC mitigates the significant drop in win rate in such cases? Additionally, how would the results vary if the flip rate were 10% or 90%?

---

> ### Author Response · Authors · 2024-11-20
> **Response to Reviewer wojJ**
>
> Thanks for your detailed and constructive review.
>
> ### Regarding more training settings (Weakness 1)
> There are several hyperparameters related to the discrete latent space: the dimension of the latent variable and the codebook size. For the dimension of the latent variable, we set it to be the same as the hidden size of the backbone LM. For the codebook size, we search in {8, 16, 32, 64, 128, 256}, and find that the performance is peaking around 32 to 64 codes (Section 4.3). As for the training of the codebook, we jointly train the codebook and other parameters of the model with the same learning rate and schedule. If you have further specific concerns regarding the training settings, we would be glad to provide more details. We will provide more thorough implementation details in the revised version.
>
> ### Regarding TSNE visualization lacking distinct clusters (Weakness 2)
> Thanks for raising this point. We want to clarify that colors in the TSNE figure represent different data sources, not different human preferences. Different data sources may have preference overlaps, so it is expected to see some overlaps in the TSNE figure. For example, we interestingly observe that data the clustering center of "truthful_qa" (colored red) is close to  "false_qa" (colored brown) while far from "flan_v2_cot" (color yellow), which can be interpreted as the preference of "truthful_qa" and "false_qa" are both related to truthfulness while "flan_v2_cot" emphasizing reasoning ability. We will add more discussions about this in the revised version.
>
> ### Regarding more evaluation datasets (Weakness 3 & 5)
> We appreciate this suggestion and will add more evaluation results on DPO as follows.
> | Method      | MMLU | MT-bench |
> | ----------- | ----------- | ----------- |
> | Llama3-8B        |  62.10      |   -     |
> | Llama3-8B w/o. LPC         | 62.68       | 6.32       |
> | Llama3-8B w. LPC         | 62.54      |  6.36      |
> | Llama3-Instruct-8B         |  63.89      |    -    |
> | Llama3-Instruct-8B w/o. LPC         | 63.60      | 7.67       |
> | Llama3-Instruct-8B w. LPC         | 63.72      |  7.82     |
>
> It is worth noting that RL fine-tuning on Ultrafeedback does not improve the performance on MMLU, which is consistent with previous works [1]. In the revision, we will provide more evaluation results on other backbone models and alignment algorithms.
>
> [1] Meng Y, Xia M, Chen D. Simpo: Simple preference optimization with a reference-free reward[J]. arXiv preprint arXiv:2405.14734, 2024.
>
>
> ### Regarding the performance of SimPO (Weakness 4)
>
> We carefully examine the evaluation setting and re-run some of the results and find that the results are consistently lower than the original SimPO paper, including the performance of the downloaded Llama3-8B-Instruct checkpoint, whose win-rate against GPT-4 is 12.90 in our evaluation and reported as 25.3 in the SimPO paper. This means that the performance gap is not caused by the implementation and training settings. We hypothesize that the performance gap is caused by the different GPT-4 version. We use GPT-4-turbo-2024-04-09 as the judge model while the SimPO paper uses  GPT-4-1106-preview. We will clarify this issue in the revised version.
>
>
> ### Regarding the issue of the flipped-label experiment (Question 2)
>
> By flipping the labels of the training data, we want to test the generalization ability of our method in the worst case where two totally opposite preferences are mixed together. By explicitly modeling the latent preference, LPC can successfully distinguish the flipped preferences from ordinary ones, outperforming standard DPO by a larger margin. Besides, we appreciate this suggestion and conduct experiments with other flipped-label rates and present the results as follows (with DPO as the alignment algorithm).
>
> |  Method      | 10% | 50% | 90% |
> | ----------- | ----------- | ----------- | ----------- |
> | Llama3-8B w/o. LPC         | 68.9      | 63.2       | 66.3      |
> | Llama3-8B w. LPC         | 70.1      | 66.2      | 67.4     |
> | diff         | +1.2      | +3.0       | +1.1  |

---

> ### Author Response · Authors · 2024-11-28
> **We are STILL looking forward to your feedback**
>
> Dear Reviewer wojJ,
>
> We have tried our best to address your concerns and respond to your questions thoroughly. At your convenience, could you please take a look at our response? We look forward to having a discussion with you.
>
> Authors of Submission 3825

---

> > ### Comment · Reviewer_wojJ · 2024-12-01
> >
> > Thank you for the response. While there are still many aspects of the experimental section that need improvement, introducing latent variables into preference alignment is an interesting attempt. Therefore, I have raised my score from 5 to 6, and I hope the author can provide more comprehensive experimental results in future versions.

---

> > > ### Author Response · Authors · 2024-12-02
> > > **Thanks for your response**
> > >
> > > Dear Reviewer wojJ,
> > >
> > > Thank you for your response. We will definitely keep improving the work based on all of the comments.
> > >
> > > Authors of Submssion 3825

---

### Official Review · Reviewer_DG67 · 2024-11-03

**Soundness:** 3
**Presentation:** 3
**Contribution:** 2
**Rating:** 5
**Confidence:** 4

**Summary:**

This paper addresses the challenge of aligning large language models (LLMs) with complex human preferences that may be conflicting across tasks and populations. The authors propose Latent Preference Coding (LPC), a framework that models implicit preference factors using discrete latent codes without relying on predefined reward functions. LPC integrates with various offline alignment algorithms and infers underlying preferences directly from data. Experiments demonstrate that LPC improves performance across multiple benchmarks and enhances robustness against noisy data, offering a promising approach for more versatile LLM alignment.

**Strengths:**

1. This paper is well written and easy to follow.
2. This paper have extensive experiments to verify the effectiveness of their proposed method.
3. The idea of using latent code to model complex human preferences is interesting.

**Weaknesses:**

1. The time computational cost of this method during training process is unclear compared to existing methods like DPO, SimPO.
2. While the idea of learning a prior distribution for latent codes is straightforward and its effectiveness has been demonstrated empirically, it is essential for the authors to further explain how this prior network can capture complex human preferences like from some theoretical proof or previous work.
3. The authors state that the advantages of latent preference coding are mitigating the notorious posterior collapse issue and achieving better interpretability. It would be beneficial to conduct an ablation study using continuous representations to learn human latent preferences.

**Questions:**

See the weaknesses above.

---

> ### Author Response · Authors · 2024-11-19
> **Response to Reviewer DG67**
>
> Thanks for your detailed review.
>
> ### Regarding time computational cost (Weakness 1)
> **The training time and training memory consumption of our method is comparable to DPO.** In our architecture design, the policy model, the prior network and the posterior network share the same backbone LM. For the input triple $<x,y_w,y_l>$, we only need to forward the backbone LM twice (once for $concat(x,y_w)$ and once for $concat(x,y_l)$), which is the same as DPO. We will provide detailed implementation details regarding the time complexity in the revised version.
>
> ### Regarding the theoretical proof or previous work of how the prior network can capture complex human preferences (Weakness 2)
> Thanks for raising this point. Using a prior distribution to model latent factors behind complex data distribution has mostly been discussed in the context of variational inference. Previous works like Variational Autoencoders (VAEs)[1] and Vector Quantized Variational Autoencoders (VQ-VAEs)[2] have shown the effectiveness of learned priors in capturing complex data distributions. In our scenario, we use a similar modeling idea to capture the complex human preferences, the feasibility of which has been validated by both empirical results and the analysis in the research direction of variational inference. We will add more related works regarding this point in the revised version.
>
> [1] Kingma D P, Welling M. Auto-encoding variational bayes[J]. arXiv preprint arXiv:1312.6114, 2013.
>
> [2] Van Den Oord A, Vinyals O. Neural discrete representation learning[J]. Advances in neural information processing systems, 2017, 30.
>
>
> ### Regarding the ablation of continuous latent representation (Weakness 3)
>
> Previous works have explored using continuous latent variables on reward modeling and found that it can successfully distinguish different human preferences [1]. Compared with continuous latent representation, discrete latent bypasses the sampling process. We choose to use discrete latent variables in our method mainly because of its simplicity of implementation and training stability. We implement a simple continuous latent representation baseline and present the results as follows.
>
> | Method      | Preference Accuracy |
> | ----------- | ----------- |
> | DPO         |  69.3      |
> | DPO w. LPC (Discrete)         | 70.8       |
> | DPO w. LPC (Continuous)         |  69.5     |
>
> We will add this ablation study in the revised version.
>
> [1] Poddar S, Wan Y, Ivison H, et al. Personalizing reinforcement learning from human feedback with variational preference learning[J]. arXiv preprint arXiv:2408.10075, 2024.

---

> ### Author Response · Authors · 2024-11-28
> **We are STILL looking forward to your feedback**
>
> Dear Reviewer DG67,
>
> We have tried our best to address your concerns and respond to your questions thoroughly. At your convenience, could you please take a look at our response? We look forward to having a discussion with you.
>
> Authors of Submission 3825

---

### Official Review · Reviewer_7o3q · 2024-11-05

**Soundness:** 2
**Presentation:** 3
**Contribution:** 2
**Rating:** 3
**Confidence:** 5

**Summary:**

This paper deals with an important problem in the alignment research: modeling of complex human preferences and proposes a method using latent discrete variables, named LPC, to solve this problem. Authors first conduct extensive objective derivations and further conduct experiments on three alignment methods: DPO, SimPO, and IPO.

**Strengths:**

1. The mathematical derivation looks fancy.
2. The paper is well-written.
3. The idea of using latent variables for various alignment objectives is intuitive and inspiring.

**Weaknesses:**

1. Although the mathematical derivation is elegant, but the experimental design and results do not demonstrate that this method is truly effective.
2. This work is like other works in past research in NLP using latent variables, presenting valid theoretical derivations but limited effectiveness, along with questionable robustness and training stability.
3. Although I really like the idea of using latent variables for various alignment objectives modeling, the architecture is too complex and seems unstable. I really worried about the practicability of this method. The authors try to prove the generalization of this method by using it on three alignment methods: DPO, SimPO, and IPO, but it is hard to ignore that the mathematical derivation is necessary for every alignment algorithms that have different alingment objective functions. This significantly undermines the practical applicability of this method. If others want to apply this method to other alignment algorithms, they would first need to perform complex mathematical derivations. Moreover, there is no way to guarantee the correctness of these mathematical derivations for other alignment algorithms.
4. The design of experiements are not good. Choosing to evaluate on the three tasks (commonsense reasoning, mathematical reasoning, truthfulness) are not convincing. I find the experiements fail to verify the effectiveness of LPC on modeling the "COMPLEX" human intentions, especially in the contradicting scenario, e.g., the scenario where we need to guarantee both helpfulness and harmlessness.
5.  The main experimental results in Table 1 are reward scores. However, the paper does not even mention what kind of reward model it is using, let alone any details that can help determine the trustworthiness of the evaluation results. If the reward model chosen in the paper is not reliable at all, the results presented in Table 1 are meaningless.

**Questions:**

1. Let's first assume the results in Table 1 is reliable. Why are some results of using LPC is much worse than not using LPC? Have you analyzed about this? Any ablation results you can share?

---

> ### Author Response · Authors · 2024-11-19
> **Response to Reviewer 7o3q (Part 1)**
>
> Thanks for your detailed review.
>
> ### Regarding Weakness 5
>
> This is a misunderstanding. **The results in Table 1 represent downstream task performance that is calculated by commonly adopted metrics (arc-challenge: acc, arc-easy: acc, gsm8k: exact_match, and truthfulQA: bleu_acc)**. LPC is mainly developed from direct preference optimization, which does not involve an explicit reward model. Besides, we do not involve any additional reward models to rate the generated responses. As a result, **our method does not produce reward scores**.
>
> ### Regarding the effectiveness of variational inference and its application in NLP (Weakness 1 & 2)
>
> We acknowledge the concern about the robustness and training stability of LPC. This is a common issue suffered by most latent-variable methods. However, it is important to note that LPC exploits discrete codes (a.k.a., vector quantization) rather than continuous variables (e.g., variables following Gaussian distributions) as latent representations of human preference. This makes a big difference, as vector quantization does not have to perform sampling from distributions, which can effectively circumvent the variance issues, as indicated by [1]. In fact, all results presented in Table 1 and Table 2 are averaged after 5 runs (each run corresponds to a different random seed). The variance of DPO w. LPC ($var_{average}=5.8e-5$) and that of DPO w/o. LPC ($var_{average}=5.1e-5$) are comparable. Detailed variance statistics for the results in Table 1 are given as follows:
>
> | Methods | Arc-challenge | Arc-easy | GSM8K | TruthfulQA | Average |
> | ----------- | ----------- | ----------- | ----------- | ----------- | ----------- |
> | Llama3-8B DPO w/o. LPC | 1.2e-4 | 1.7e-5 | 4.5e-5 | 5.3e-5  | 5.8e-5 |
> | Llama3-8B DPO w. LPC | 4.2e-5 | 4.9e-5 | 4.1e-5 | 7.4e-5 | 5.1e-5 |
> | Llama3-8B SimPO w/o. LPC| 8.2e-5 | 4.1e-5 | 1.8e-4 | 3.4e-5 | 8.4e-5 |
> | Llama3-8B SimPO w. LPC| 6.9e-5 | 4.4e-5 | 1.3e-4 |  3.3e-5 | 6.9e-5 |
>
>  To make our results more convincing, we report the preference accuracy of the 5 runs as follows:
>
> | Methods | run 1 | run 2 | run 3 | run 4 | run 5 | Average |
> | ----------- | ----------- | ----------- | ----------- | ----------- | ----------- | ----------- |
> | Llama3-8B DPO w/o. LPC | 69.1 | 69.3 | 69.4 | 69.4 | 69.3 | 69.3 |
> | Llama3-8B DPO w. LPC| 70.2 | 70.8 | 70.6 | 71.2 | 70.7 | 70.8 |
> | Llama3-8B SimPO w/o. LPC| 69.6 | 69.2 | 69.4 | 68.7 | 69.4 | 69.2 |
> | Llama3-8B SimPO w. LPC| 72.3 | 71.4 | 71.7 | 72.7 | 70.4 | 71.8 |
>
> From the detailed results above, we observe stable and consistent improvements. We hope these details (as well as the clarification above for Table 1) can convince you regarding the effectiveness and stability of the proposed method.
>
> We understand that there might be different opinions toward latent-variable models. Such methods have been well-established in computer vision [1][2], and attain a lot of interesting progress in NLP, including some recent achievements [3][4]. Then, can we say the exploration on alignment with latent variables is invalid, just because of the tags in our mind for latent variables based on ``past NLP research”? In that case, we would not see the prosperity of PPO in LLM alignment, because for a long time, the community of NLP think RL does not work for NLP [5]. Echoing your comment, we also think variational inference is suitable, intuitive, and rational for preference learning, yet detailed methodology has not been well developed. This motivates the work. We hope our effort can inspire more discussions and studies about variational inference for preference modeling in the NLP community. We also hope the research community could be more open-minded and inclusive, encouraging diverse approaches and explorations.
>
> [1] Aaron van den Oord et al., Neural Discrete Representation Learning. In NeurIPS 2017.
>
> [2] D.P. Kingma and Max Welling, Auto-Encoding Variational Bayes. In ICLR 2014. (Test of Time Award on ICLR 2024)
>
> [3] Aurko Roy and David Grangier. Unsupervised Paraphrasing without Translation. In ACL 2019
>
> [4] Lin Y. C., et al., ACCEPT: Adaptive Codebook for Composite and Efficient Prompt Tuning. Findings of EMNLP 2024.
>
> [5] Rajkumar Ramamurthy et al., Is Reinforcement Learning (NOT) for natural language processing: benchmarks, baselines, and building blocks for natural language policy optimization. In ICLR 2023

---

> ### Author Response · Authors · 2024-11-19
> **Response to Reviewer 7o3q (Part 2)**
>
> ### Regarding practical applicability of LPC (Weakness 3)
> While the mathematical derivation of LPC is mainly based on DPO in the draft,  LPC is a general framework that can be extended to other alignment algorithms. The purpose of the derivation is to provide an intuitive explanation of how LPC modifies the single-preference alignment algorithm. The key insight behind the derivation is that the preference model $p(y_w ≻ y_l | x, z)$ can be represented as $\mathbb{E}_{z \sim p(z|x)}[p(y_w ≻ y_l | x, z)]$, which introduces a new variable $z$ to model the latent preference. As long as the alignment algorithm optimizes the log-likelihood of $p(y_w ≻ y_l | x, z)$ (as far as we know, this covers most popular alignment algorithms), LPC can be applied to these methods with strict mathematical derivation. As a result, after the comprehensive derivation on DPO, one can make a simple analogy for other alignment algorithms without much heavy derivations, just like what we did in Section 3.4. In the revision, we will reorganize the derivation part to make it clearer.
>
> ### Regarding Weakness 4
> We conduct additional experiments on a scenario with complex human intentions to validate the effectiveness of our method. We take the helpfulness and truthfulness labels in the Ultrafeedback dataset as two preference directions and construct a new dataset whose preference scores are mixed between the two directions. Specifically, for each instance, we construct $<y_w, y_l>$ pair using the two directions with equal probability. We find that our method can still achieve a satisfying performance, which indicates the effectiveness of our method in handling complex human intentions. The preference accuracy is as follows:
>
> | Method      | Truthfulness vs. Helpfulness | Helpfulness vs. Honesty |
> | ----------- | ----------- | ----------- |
> | LLama3-8B DPO w/o. LPC         |  62.2/64.5       | 67.6/65.1
> | LLama3-8B DPO w. LPC         |  63.8/65.0       | 68.8/65.5
>
> From the results, we can see that LPC brings a performance gain on both two preferences, indicating the effectiveness of our method in handling complex human intentions. We will add a more thorough version of this experiment in the revision.
>
> ### Regarding performance issue (Question 1)
> According to the results in Table 1, the largest performance drop of using LPC is 1.62 (training SimPO on Llama3-8B, evaluating on Arc-challenge), which is debatable to be perceived as being "much worse" than not using LPC. On the contrary, LPC brings good performance gain on most of the tasks and models (up to 14.08, training SimPO on LLama3-8B, evaluating on Arc-challenge). We also observe that these minor performance drops mainly occur in SimPO. For DPO, LPC consistently brings performance gains.

---

> ### Author Response · Authors · 2024-11-28
> **We are STILL looking forward to your feedback**
>
> Dear Reviewer 7o3q,
>
> We have made every effort to address your concerns and respond to your questions thoroughly. At your convenience, could you please take a look at our response? We look forward to having a discussion with you.
>
> Authors of Submission 3825

---

### Meta-Review · Area_Chair_P2Hs · 2024-12-23

**Metareview:**

This is a paper that studies alignment and seeks to tackle one of the main challenges—the fact that human preferences are not consistent but are rather the result of a collection of underlying factors and origins. The way the authors tackle this issue by proposing a latent codebook in alignment. They then integrate an approach to learn the latent variable within standard alignment techniques like DPO.

The paper has a really nice idea overall and the writing is very clear. The main element that is missing here is consistently strong results. The authors mainly focus on a number of standard benchmarks. The results show some improvement, but it is difficult to validate whether the proposed approach is really doing the work here, or whether something else is going on. What would be better, and what I would suggest for the next iteration of the paper, is to start with a fully synthetic setting where there are multiple latent preferences that are known (and controllable) and where it is possible to fully test out the algorithm. The authors do a little bit of this in their label-flipping heuristic, but this was limited. Instead they could start with multiple synthetic reward functions that have controllable distributional distance, and use these to generate their data.

For the real-world experiments, the bigger question is just how much these are aligned with the authors’ model, and this is pretty difficult to tell from the paper.

Overall, this is a great start to a paper that needs just a bit more work and will soon be quite strong. I believe it will benefit from one more submission cycle.

**Additional Comments On Reviewer Discussion:**

Most of the reviewers asked questions around complexity, practical training considerations, etc. The authors did a good job responding to these. The rebuttal increased my opinion of this paper, but I still felt there was a little bit more work to do to clear the bar.

---

### Decision · Program_Chairs · 2025-01-22

Reject